# Use of a Collar-Mounted Triaxial Accelerometer to Predict Speed and Gait in Dogs

**DOI:** 10.3390/ani11051262

**Published:** 2021-04-27

**Authors:** Samantha Bolton, Nick Cave, Naomi Cogger, G. R. Colborne

**Affiliations:** School of Veterinary Science, Massey University, Palmerston North 4442, New Zealand; sammybolton96@gmail.com (S.B.); n.cogger@massey.ac.nz (N.C.); grcolbor@gmail.com (G.R.C.)

**Keywords:** accelerometry, treadmill, speed, gait, prediction, dog

## Abstract

**Simple Summary:**

Accelerometers have been used for several years to monitor activity in free-moving dogs. The technique has particular utility for measuring the efficacy of treatments for osteoarthritis when changes to movement need to be monitored over extended periods. While collar-mounted accelerometer measures are precise, they are difficult to express in widely understood terms, such as gait or speed. This study aimed to determine whether measurements from a collar-mounted accelerometer made while a dog was on a treadmill could be converted to an estimate of speed or gait. We found that gait could be separated into two categories—walking and faster than walking (i.e., trot or canter)—but we could not further separate the non-walking gaits. Speed could be estimated but was inaccurate when speed exceeded 3 m/s. We conclude that collar-mounted accelerometers only allowing limited categorisation of activity are still of value for monitoring activity in dogs.

**Abstract:**

Accelerometry has been used to measure treatment efficacy in dogs with osteoarthritis, although interpretation is difficult. Simplification of the output into speed or gait categories could simplify interpretation. We aimed to determine whether collar-mounted accelerometry could estimate the speed and categorise dogs’ gait on a treadmill. Eight Huntaway dogs were fitted with a triaxial accelerometer and then recorded using high-speed video on a treadmill at a slow and fast walk, trot, and canter. The accelerometer data (delta-G) was aligned with the video data and records of the treadmill speed and gait. Mixed linear and logistic regression models that included delta-G and a term accounting for the dogs’ skeletal sizes were used to predict speed and gait, respectively, from the accelerometer signal. Gait could be categorised (pseudo-R2 = 0.87) into binary categories of walking and faster (trot or canter), but not into the separate faster gaits. The estimation of speed above 3 m/s was inaccurate, though it is not clear whether that inaccuracy was due to the sampling frequency of the particular device, or whether that is an inherent limitation of collar-mounted accelerometers in dogs. Thus, collar-mounted accelerometry can reliably categorise dogs’ gaits into two categories, but finer gait descriptions or speed estimates require individual dog modelling and validation. Nonetheless, this accelerometry method could improve the use of accelerometry to detect treatment effects in osteoarthritis by allowing the selection of periods of activity that are most affected by treatment.

## 1. Introduction

Accelerometry has the potential to be a powerful tool for investigating activity in dogs. It has been used to assess the efficacy of treatment in dogs with various diseases, most notably in osteoarthritis (OA) [1,2,3].

The most commonly used unit of measure from commercial accelerometer systems is an activity count, an arbitrary measure of movement intensity [1,2]. It is derived by summing the acceleration across the three planes (delta-G), after proprietary data cleaning and amalgamation processes [4]. As a consequence of company-specific methods of data manipulation, the activity count for a given delta-G can vary between accelerometer systems.

Activity counts have been used to monitor total night-time activity, total daytime activity, and increased intensity of activity to evaluate the effectiveness of several OA treatments, such as green-lipped mussel extract nutraceuticals [1,2,3].

There is potential to use accelerometry to produce measurements beyond the basic activity count. Triaxial accelerometers work by continuously recording acceleration across three orthogonal planes. When these planes are orientated with the sagittal, coronal, and transverse planes of a dog’s body, the resultant acceleration measurements may be able to characterise activity and gait in dogs. Accelerometry has not yet been used to estimate speed in free-moving dogs. However, in one study, the authors were able to identify significant differences in the accelerometry data at different speeds [5]. The study only used two different speeds, and therefore the differences in accelerometry output could have been due to differences in gait and not specific to speed.

Accelerometry has been used to detect categories of activity. Several studies have used accelerometry to determine whether a dog was walking, sleeping, cantering, or in a period of “inactivity” [6,7,8]. For activity categorisation, algorithms have been created to identify patterns in the acceleration vectors consistent with each activity in question, such as the symmetrical gait of a trot [6,7,8]. This method has been shown to accurately identify certain activities in dogs. However, categorisation of activities using the activity count to characterise activity does not appear to have been investigated. One reason for this is probably that the accuracy of categorisation using this method may be too low. In addition, it is easier and simpler to use the activity count rather than to apply complex algorithms to the continuous data, especially for long-term studies, such as those required to test the efficacy of nutraceuticals in the management of OA. Nonetheless, it remains to be seen whether gait categorisation offers any advantage.

Categorising locomotion by gait or speed is more complex and requires more data transformation than that required to produce the activity count, but there are significant benefits to doing so. For reporting purposes, delta-G has little meaning to people without accelerometry experience. Transforming the data into a parameter most people readily understand, like gait or speed, would allow for the effective communication of experimental results. It would also allow the division of periods of continuous activity data into periods of time more likely to contain evidence of disease. For instance, lameness is more readily detected in dogs while trotting than while walking [9], and lame dogs would be more likely to walk than trot.

Translating the activity count into a clearly defined variable such as speed or gait categories avoids using arbitrary cut-off points within the activity count data. Converting accelerometry data into defined gaits and speeds could improve our ability to detect changes in locomotion, which could be used to assess lameness and response to treatment. If successful, it would improve the use of accelerometry to detect treatment effects in OA by allowing the selection of the key activities or speeds that are most affected by treatment. Therefore, this study aimed to measure controlled activity in dogs using accelerometry and determine whether the “delta-G” measures of activity (as opposed to a propriety specific activity count) could be reliably converted to estimates of speed and categorisation of gait.

## 2. Materials and Methods

### 2.1. Animal Selection

Eight Huntaway dogs were selected from a population held at Massey University’s Canine Colony. All dogs were between three and 10 years of age and deemed healthy by a veterinary physical examination before study commencement. Due to the novel nature of this study, the sample size required was unable to be calculated a priori based on previous studies. However, the most comparable study used 6 dogs to identify optimal accelerometer placement for broad activity categories on a treadmill [5]. Therefore, it was estimated that eight dogs would provide enough data to validate the use of delta-G applying our proposed method. This study was approved by the Massey University Animal Ethics Committee (Protocol 18/44).

### 2.2. Accelerometer

A micro electro-mechanical (MEMS) triaxial accelerometer (Heyrex^®^, Say Systems, Wellington, New Zeland), weighing 32 g and measuring 65 × 26 × 18 mm, was used for this study (Figure 1). The collar-mounted accelerometer was positioned on the ventral aspect of each dog’s neck. Accelerations between +4 and −4 G in magnitude were recorded at a sampling rate of 10 Hz. Delta-G was calculated across three axes as the change in acceleration between contiguous samples and summed into one-second epochs. When in the range of the proprietary receiver, the accelerometer telemetered captured acceleration data to proprietary software for cleaning, transformation, and summation.

### 2.3. Experimental Procedure

The dogs were acclimated to the treadmill and safety harness over a six-month period. During the acclimation period, the dogs were trained up to three times a week until they could confidently move on the treadmill without excessive interference from handlers. The treadmill belt was 8.36 m in length and was set without an incline for both the acclimation period and experiment. The speed of the treadmill was calibrated by the manufacturer, shortly before study commencement. In addition, the speed was checked at the time, using markers that were secured to the belt at a precisely measured distance apart, and their sequential movement was timed as they passed a static point at different speeds. The safety harness was a vest worn by the dog (Figure 1) that was attached by ropes to the frame beside the treadmill. The ropes prevented the dog from accelerating beyond the speed of the treadmill and supported the dog until someone stopped the treadmill if it stumbled. Otherwise, the ropes were loose enough not to impede the dog’s gait.

On the day of data capture, all dogs were fitted with the same adjustable safety harness, an accelerometer was attached to their collar, and retroreflective markers were placed on the dog (see video analysis for more detail), as shown in Figure 1. Each dog was led onto the treadmill and encouraged to move at a slow walk, fast walk, slow trot, fast trot, slow canter and fast canter for 30 to 50 s in each gait. For each gait, periods of 10 s were recorded using the motion capture software, and treadmill speed was recorded. The recordings were aligned to the measure of delta-G captured by the accelerometer by a precise timestamp with the annotated motion capture. The order that dogs completed each gait differed based on the dogs’ preferences on the day. For example, some dogs were excited and willing to canter when first placed on the treadmill, while others appeared more comfortable initially walking. The treadmill’s speed at each gait was set to a speed that allowed each dog to move comfortably for 30 to 50 s without changing their gait. It was impossible to record all gaits in some dogs because they would not walk or canter. Periods were discarded from analysis for an inconsistent gait, obscuration of the markers, and excessive forwards/backwards movement of the dog that occurred because the dog was moving faster or slower than the set treadmill speed.

### 2.4. Video Analysis

Six infrared cameras were placed around the treadmill to capture a three-dimensional view of retroreflective markers placed on the dog. The calibrated volume of space was approximately 4 m in length, 1 m in width, and 1 m in height, and calibration was performed with a static L-shaped frame on the treadmill belt and a wand with two markers spaced 0.6 m apart. Flat markers (1 cm diameter) were placed on the metacarpophalangeal and metatarsophalangeal joints and spherical markers (1 cm diameter) on the dorsal midline of the harness between the scapulae and on the lumbosacral joint (Figure 1). The motion captured by the marker movement was converted into calibrated movement in the vertical, horizontal, and transverse axes with a proprietary software system (Qualisys Track Manager v. 2.17, Gothenburg, Sweden). The movement of the foot markers was used for visual gait identification and for identification of stride duration as defined by Nunamaker and Blauner [10].

### 2.5. Morphometry and Estimation of Skeletal Size

We wished to determine whether predictive models would be improved if we accounted for the dog’s skeletal size. It is unknown whether a single skeletal measurement, such as limb length, might substitute for all skeletal measurements such as height, body length, or thoracic girth, and which would not be overly affected by differences in conformation between dogs. Thus, we used a composite variable, termed “skeletal size”, which was generated using principal component analysis, to reduce several measurements to a single value, the principal component. The approach to use PCA on morphometric measurements to create the single composite expression of skeletal size has been previously used in a study of New Zealand Huntaways [11].

Briefly, six skeletal measurements were taken from each dog between specific bony locations using a flexible tailor’s measuring tape as described in Table 1. For each of the six skeletal measurements, the values from the eight dogs were combined using the factor loadings as described by Leung et al. [11] (Table 1). The factor loadings from the first principle component for each skeletal measure were used to generate the singular variable for each dog (“skeletal size”), which is considered to account for the majority of variation between the measurements. For a full description of the approach, the reader is referred to the paper by Leung et al. [11].

### 2.6. Statistics

Statistical analysis was performed using the statistical software R (version 3.5.2, R Development Core Team; R Foundation for Statistical Computing, Vienna, Austria). Descriptive statistics and the distribution of variables were investigated. The variables were delta-G, speed, gait, body weight, age, and skeletal size. Pearson’s correlation coefficient was used to evaluate the relationship between speed and delta-G. Delta-G was defined as the change in acceleration between adjacent 0.3 s sampling time points summed across three axes.

Separate multivariate models were constructed to determine whether delta-G could be used to predict speed or gait. Speed was expressed in meters per second, and initially, gait was an ordinal variable with three levels. However, an examination of the data did not identify a convincing break between delta-G values for trotting and cantering, and for different speeds. Consequently, gait was compressed into a two-level binary variable, called “binary gait variable” (BGV), with two levels that were coded 0 if the dog was walking, and 1 if the dog was trotting or cantering.

A mixed-effects linear regression model was constructed to predict speed based on delta-G, in addition to body weight, age, and skeletal size. A forward and backward stepwise process was used for selection of fixed effects, which started with a full model and eliminated variables one at a time, and then the eliminated variables were returned into the reduced model to ensure the model was not improved, before removing variables again using the stepwise function in the Mass package in R. The model was then extended to include a random effect for dog, to account for both the repeated measures, and dog-specific gait characteristics unaccounted for in skeletal size. The goodness of fit of the model was assessed by comparing the AIC and log-likelihood between the mixed model and a mixed-effects intercept-only model. The model assumption of independence was handled with the inclusion of dog as a random effect, the assumption of normality of residuals was checked visually with a Q–Q plot and equal variance of residuals was assessed with a plot of residuals against fitted values. The assumption of a linear relationship between the outcome variable and the predictor was assessed by the addition of a quadratic into the model.

A mixed-effects logistic regression model was constructed to predict gait using the two-level categorical variable BGV, and any additional explanatory variables that significantly added to the model’s predictive ability. A stepwise selection process was used to select final fixed effects in the model as described previously in this section. The model was then extended to include a random effect for dog to account for the repeated measures. The goodness of fit of the model was assessed by comparing the log-likelihood ratio statistic and the AIC probability, deviance and AIC of the full mixed model with the mixed-effects intercept-only model. Hosmer and Lemeshow’s pseudo-R2 was also calculated for the simple logistic regression [12]. The model assumption of scale—that is that the relationship between speed and delta-G on the logit scale was linear—was tested with the inclusion of a quadratic of the delta-G term into the model.

## 3. Results

Eight dogs were enrolled in this study. However, prior to analysis, one dog was removed from this study due to failure to persist in a singular gait for a 10 s period on the treadmill. Of the seven dogs included in the analysis, six were female and one was male. Body weight ranged from 21 to 25.8 kg. The first principal component accounted for 56% of the variation in morphometric measurements between dogs. The eigenvalue for body length was much smaller than the values for the other measurements, indicating that it had less influence on overall skeletal size than the other measurements (Table 1). The relationship between skeletal size and body weight for the seven dogs that contributed data to this study is shown in Figure 2.

The seven dogs contributed 345 discrete measurements of delta-G summed over 10 s intervals (delta-G10 s). Of those, 34 intervals were removed due to inconsistent gait or speed. The remaining dataset included 311 measurements of delta-G10 s: 113 were recorded when the dogs were walking, and 198 were recorded when the dogs were either trotting or cantering. Speed ranged from 0.67 m/s to 6.87 m/s. The distribution of delta-G10 s overall speeds was bimodal (Figure 3). The relationship between speed and delta-G for all the data points is shown in Figure 4, while Figure 5 shows the same relationship for each dog.

There was a strong relationship between speed and delta-G for the whole dataset, with a Pearson’s correlation coefficient of 0.89. Visual assessment of this relationship indicated that the relationship was linear for each individual dog, though it appeared non-linear as a whole dataset (Figure 4 and Figure 5). Including “dog” in the model as a random effect accounted for this phenomenon, improving the model and negating the need for data transformation.

As stated previously, there was a division in the delta-G-speed curves that corresponded to the change in gait between walking and trotting, but not between trotting and cantering. The interquartile range of delta-G for the walking intervals (min: 14.23, LQ: 23.59, median: 29.5, mean: 33.71, UQ: 42.32, max: 64.42) did not overlap with the trotting/cantering IQR (min: 31.70, LQ: 97.42, median: 122.10, mean: 119.86, UQ: 139.85, max: 205.00). In contrast, there was significant overlap between the delta-G values for dogs while trotting and cantering, which led to the binary categorisation of BGV, rather than a tertiary categorisation of their actual gait.

The final mixed-effects linear regression model to predict speed included delta-G and skeletal size as fixed effects after stepwise selection, and dog as a random effect (Table 2). The F-statistic and effect size of the model without the random effect of dog were both significantly high (F2, 308 = 694.5, *p* < 0.001; R2 = 82%), and both explanatory variables were significant, indicating that the model was a good fit of the data. The model was then checked in its full form, with dog added back in. When compared with the intercept-only mixed model, the mixed model fitted better (Table 2). All model assumptions were met except one, which was that there was not equal variance of residuals, whereby the plot of residuals against fitted values of speed produced a clear funnelling trend (Figure 6).

The final mixed-effects logistic regression model constructed to predict BGV contained only delta-G as a fixed effect after stepwise selection of variables, and dog as a random effect (Table 3). Comparison of the full model with the intercept-only mixed model supported the full model as a better fit of the data, with a significant log-likelihood ratio statistic (χ^2^ (1) = 35403, *p* < 0.001). The Hosmer and Lemeshow’s pseudo-R2 was high at 87%, indicating that the fit of the model improved greatly with inclusion of the delta-G variable. All model assumptions were met.

## 4. Discussion

This study explored whether the change in delta-G measured using a triaxial accelerometer could be used to predict speed and gait of dogs running on a treadmill. Analysis of gait showed that the odds of a dog trotting or cantering increased with delta-G, and the model was a good fit for the prediction of the BGV categorisation of a dog on the treadmill. Gait was categorised into the binary division of walking, and a gait faster than walking, because a clear cut-off point was identified in the data. It was not possible to differentiate the faster gaits into trotting and cantering because there was no clear cut-off point for delta-G values (Figure 4). While there was a clear distinction between trotting and cantering for some dogs, suggesting that it is possible to use this method for gait classification of individual dogs, the demarcation point varied between dogs considerably. Therefore, it would be required that setpoints were established for every individual dog—a method that cannot be applied to an undefined population of dogs. Individual dogs may increase their speed by increasing stride frequency preferentially or earlier than other dogs, who may use stride length instead. As the processed signals may depend on stride frequency, this could introduce some variability between dogs. However, further gait categorisation may not be necessary when the intention is to identify periods of activity when a dog is travelling fast enough to highlight injury, illness, or a therapeutic effect, as was our intent.

The confidence of binary gait classification for reporting purposes is high, which is promising, as, like speed, gait is a well-understood classification, and the model built in this study has a very good fit. An important function of gait classification beyond reporting is for identifying periods of activity of interest within the delta-G data, so that large datasets can be filtered for easier and more precise analysis. As previously mentioned, clinical signs of disease such as OA, are more likely to be detected during periods of greater activity intensity [9]. Given the model fit, we have a high degree of confidence in identifying periods when a dog is moving at speeds faster than walk. Other methods that can reliably differentiate more gaits have used a more complex method of categorisation, with algorithms to identify patterns in the accelerometry beyond simple delta-G analysis or activity count [6,7,8]. This indicates a more complex method than delta-G analysis may be required for a more detailed categorisation of gait. The disadvantage of such an approach is the need for more extensive data processing, which is very difficult for long-term studies on several individuals due to the size of the datasets generated.

In the mixed-effects model (Table 2), the dogs’ predicted speed increased by 0.3 m/s for each 10 unit increase of delta-G, when accounting for skeletal size and the repeated measures of the dogs. This study’s findings agree with those by Preston et al. [5], who reported that accelerometer vector magnitude (similar to an activity count as defined in the introduction) can be used to identify a change in speed. The inclusion of skeletal size improved the model’s predictive ability significantly, which indicates that in order to determine speed using data from the accelerometer used in this study, the dog’s size must also be taken into account. The model presented here capably identifies a change in speed, but the accuracy of the estimation of the absolute speed value is low. Therefore, its use is limited to detecting differences in speed between walk and the faster gaits, rather than estimating absolute values. The desire to estimate speed from the accelerometry data arose because speed is a term understood by most individuals, whereas delta-G is non-intuitive, and quantitatively meaningless without considerable experience, context, and comparative data. Unfortunately, the accuracy of speed estimation decreased with increasing speed, as depicted by the model residuals. Again, possibly due to the trade-off between increasing speed by stride length or stride frequency. It remains to be seen whether rough estimates with large confidence intervals are still useful for communication, or detection of treatment effects in dogs with OA.

For the linear mixed regression model, the assumption of the equal variance of residuals was violated. The impact of this violation is on the estimated standard error for the beta coefficients of explanatory variables, where the standard error is possibly underestimated. However, it is not believed that there is a significant impact on the estimation of the coefficient. Therefore, while it is acknowledged that there is a violation of the model assumption, the model was used for the remainder of this study.

A major limitation of this study was the controlled environment in which it was carried out. Each interval was recorded on a treadmill at a constant speed and incline for a defined time period, which has implications for the use of both models to predict speed or gait in a free-living environment. It is unlikely that a dog running free will remain within a consistent gait and uniform speed for a 10 s period that aligns with the epoch defined by this particular accelerometer. It is far more likely that any given 10 s period of consistent activity straddles two epochs, or perhaps, particularly for high-intensity activity, a dog may not carry out the activity for a complete 10 s period at all. Therefore, inference of results must acknowledge that the averaged activity across the epoch may not reflect the true activity of a dog, and speed may be underestimated, or gait misclassified. It remains to be seen whether collection of sufficient data over a long enough trial period overcomes these inaccuracies.

Although not specifically studied in dogs, the effect of a dog gaiting on a treadmill as opposed to the ground is expected to be negligible on the accelerometry output despite the known differences in gait between the two. This is supported by evidence of insignificant differences in activity counts per minute for humans on and off a treadmill while walking and running, supporting the use of models built with treadmill data for use on land data [13,14]. Additionally, in horses, the effect of treadmill locomotion on back kinematics in comparison to over-ground locomotion was negligible using motion capture software [15]. For this reason, it is likely that the accelerometer output between treadmill and ground running would be similar in dogs also. Nonetheless, running over uneven or inclined ground may produce differences in the delta-G at a given speed that leads to gait miscategorisation or further inaccuracy in speed estimates.

Another important limitation of this study is that neither of the models built have been validated. To validate the use of these models for predicting the speed and gait of dogs on a treadmill, another set of data would need to be obtained, against which these models were applied. Comparison of the predicted values against the true values of this new dataset would reveal the usability of these models. As the small sample of seven dogs used in this study were the same breed, of similar weight and size, and almost all were female, in order to validate these models for application to dogs in general, a far more diverse range of dogs would need to be used, if inference beyond this narrow dog type is desired. Similarly, the validation of these models for predicting the speed and gait of free-moving dogs, would require simultaneous speed and accelerometry measurements in dogs running free. That would be difficult, since the measurement of speed would require the use of a speed radar, time/distance markers, or similar. It would also be difficult to obtain 10 s of continuous gait or speed without human intervention.

## 5. Conclusions

In conclusion, this study has demonstrated that delta-G can be used to separate a dog’s gait into walking and a faster gait (trot or canter), but estimation of speed from the accelerometer signal above 3 m/s was inaccurate. We were unable to determine whether that inaccuracy was due to the sampling frequency of the particular device used, or whether that is an inherent limitation of collar-mounted accelerometers in dogs. While the model has not been validated, there is still value in using the model to screen large datasets in the field, to subset those 10 s epochs that involve movement at a gait faster than a walk, in order to provide an objective measure of therapies in dogs with OA. In contrast, the model to predict speed would benefit from further revisions if it was considered necessary to generate accurate standard errors. However, even with this limitation, the model may be of use for the identification of changes in speed between epochs rather than absolute speed measures.

## Figures and Tables

**Figure 1 animals-11-01262-f001:**
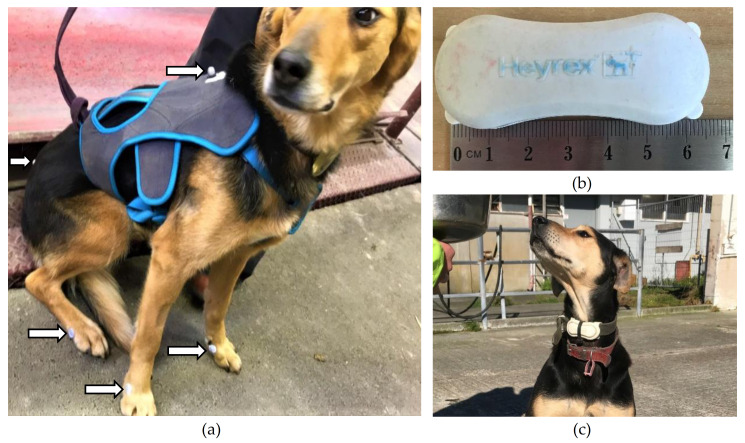
(**a**) indicates the placement of markers (white arrows) on each dog for motion capture of the dog as it moved on the treadmill, (**b**) is of the accelerometer used in this study, (**c**) is of the accelerometer on the dog, attached to the collar and positioned on the ventral side of the neck.

**Figure 2 animals-11-01262-f002:**
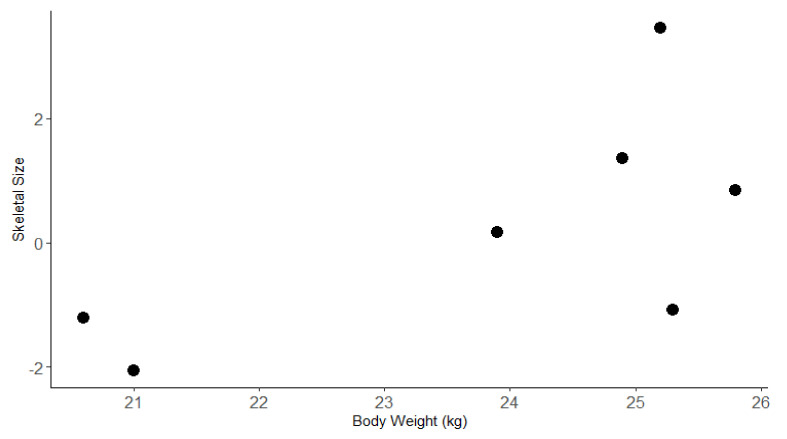
Plot of skeletal size variable against the bodyweight of the seven participating Huntaway dogs. Skeletal size was a variable calculated with the eigenvalues from six morphometric measurements taken of the body.

**Figure 3 animals-11-01262-f003:**
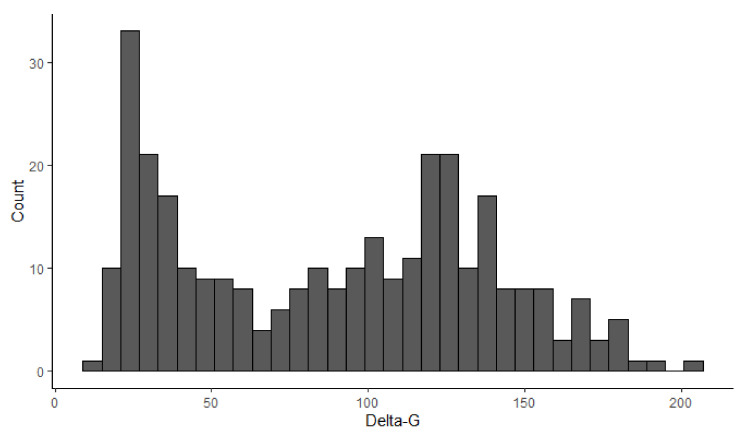
Distribution of 311 delta-G values recorded with a collar-mounted accelerometer for 10 s intervals of seven dogs at different speeds on a treadmill (mean: 88.70, median: 94.59, min: 14.23, max: 205.00, range: 190.77, LQ: 37.39, UQ: 126.89, IQR: 89.5).

**Figure 4 animals-11-01262-f004:**
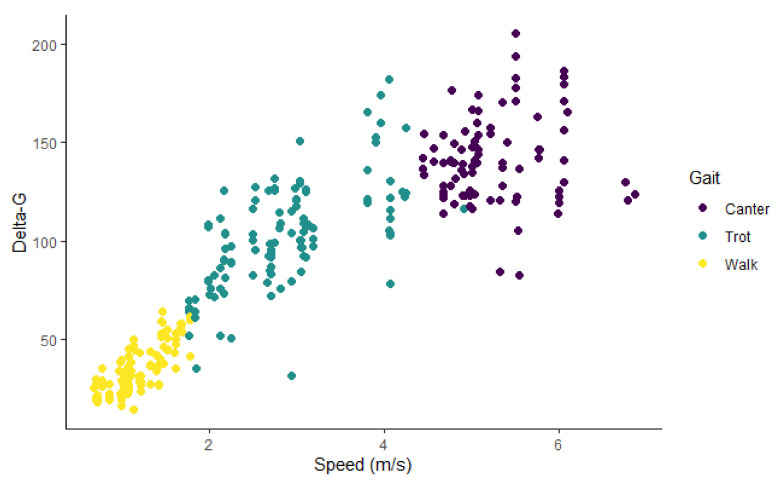
Plot of delta-G and speed with the distribution of the three gaits, walk, trot and canter. Data from 311, 10 s intervals of seven Huntaway dogs on a treadmill.

**Figure 5 animals-11-01262-f005:**
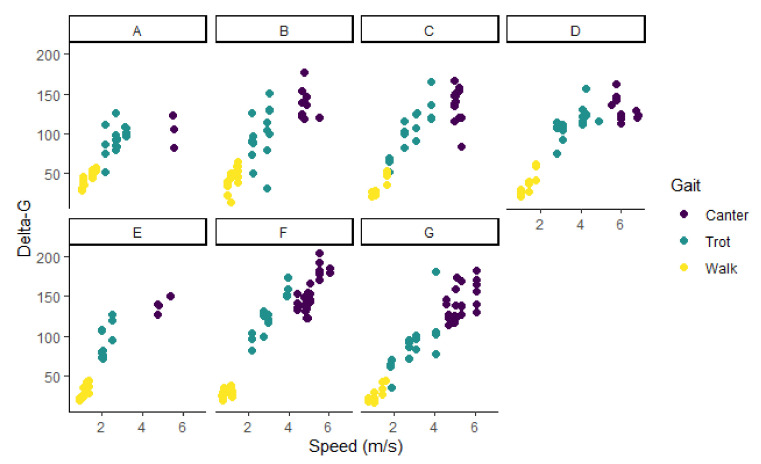
Scatterplot of speed and delta-G, and the distribution of gait for seven Huntaway dogs (A to G) on a treadmill.

**Figure 6 animals-11-01262-f006:**
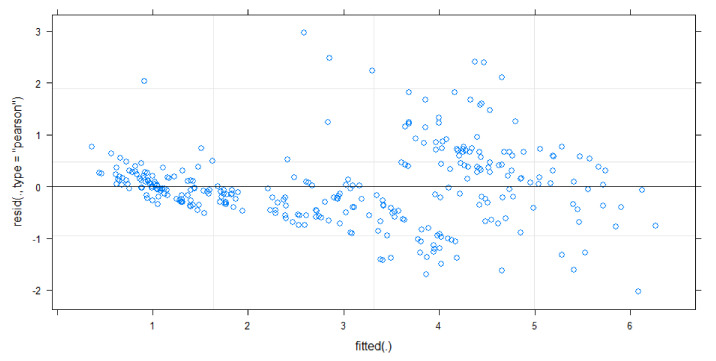
Plot of residuals against the fitted speed values for the linear mixed model predicting speed with delta-G and skeletal size, with dog as a random effect.

**Table 1 animals-11-01262-t001:** Description of the six morphometric measurements (table is taken from Leung et al. [11]), in addition to the mean morphometric measurements and loadings for the first principal component from the eight Huntaway dogs enrolled in this study.

Measurement	Description	Mean (cm)	1st PC
Head circumference	Circumference at the widest point, equidistant between the eyes and ears	39.56	0.331
Head length measurement	Distance from the level of the medial canthi, equidistant between the eyes, to the external occipital protuberance	13.19	0.387
Thoracic girth	Chest circumference at the level of the xiphoid process	70.06	0.481
Body length	Distance from the dorsal process of thoracic vertebra 1 (T1) to the dorsal process of sacral vertebrae 1 (S1)	44.62	0.190
Foreleg measurement	Distance from the proximal edge of the central foot pad to the olecranon process	27.31	0.477
Hind-leg measurement	Distance from the proximal edge of the central foot pad to the dorsal tip of the calcaneal process with the tarsus in extension	14.25	0.496

**Table 2 animals-11-01262-t002:** Results from a mixed-effects linear regression model predicting speed with repeat measurements in individual dog accounted for with an intercept-only random effect.

	Beta	SE Beta	95% CI	*p*-Value
Intercept	0.104	0.099	−0.082, 0.290	
Delta-G	0.031	0.001	0.030, 0.033	<0.001
Skeletal Size	0.137	0.037	0.067, 0.207	<0.001
Dog-level random effect				0.25 ^a^

**Table 3 animals-11-01262-t003:** Results from a mixed-effects logistic regression model predicting BGV with delta-G, with repeat measurements in individual dog accounted for with an intercept-only random effect.

	Beta (SE)	*p*-Value	95% CI for Odds Ratio		
			Lower	Estimated Odds Ratio	Upper
Intercept	−10.238 (1.849)	<0.001	9.536 × 10^−7^	3.577 × 10^−5^	1.341 × 10^−3^
Delta-G	0.170 (0.031)	<0.001	1.115	1.185	1.261
Dog-level random effect		0.550 ^a^			

Deviance of 53 with 3 degrees of freedom. The between dog variance (intercept) was 0.473. ^a^
*p*-value calculated using the likelihood ratio test.

## Data Availability

Data for this study can be found in the Appendix A as the “treadtrial_2021.csv”.

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
