# Peer review of "Use of a Collar-Mounted Triaxial Accelerometer to Predict Speed and Gait in Dogs"

_animals, 2021, doi:10.3390/ani11051262_

Round 1

Reviewer 1 Report

This is a well written and presented article.  I only have very minor comments.

Please check the use of the word "if."  For example, Abstract line 20.  "Whether" would be appropriate.  This occurs multiple times throughout, and is used correctly in Discussion, line 280.

Please be consistent with "categorise" verses "categorize."

The use of color in Figures 4 and 5 would be helpful.

Author Response

Reviewer #1

This is a well written and presented article.  I only have very minor comments.

Please check the use of the word "if."  For example, Abstract line 20.  "Whether" would be appropriate.  This occurs multiple times throughout, and is used correctly in Discussion, line 280.

Answer: We’d like to thank the reviewer for their attention to detail.  With indirect questions such as in the abstract, I have always used “if” and “whether” interchangeably, and only preferred “whether” when it is in a sentence with “or”.  In the abstract, I judge the usage to be appropriate, though in lines 58, 177, and 332 it is agreed that “whether” is more appropriate, and the manuscript has been changed accordingly.

Please be consistent with "categorise" verses "categorize."

Answer:  We apologise for the inconsistency, and have chosen “categorise” for the manuscript.

The use of color in Figures 4 and 5 would be helpful.

Answer:  The decision to use grey-scale was to pander to the crippling combination of colour blindness and ignorance by one of the authors (Cave)!  We have changed it back to colours that we believe are more suitable.

Reviewer 2 Report

[Animals] Manuscript ID: animals-1182843

Thank you for submitting an interesting and important manuscript. The manuscript deals with outcome measures and the development of treatment methods to be used in animal physiotherapy, canine physical rehabilitation and sports medicine. Such clinically relevant articles are very much needed.

The authors have obviously put a lot of consideration into relevant factors concerning physical activity and canine locomotion, which has resulted in a well-thought-out and interesting submission. This study is important and I want to encourage the authors to put effort into revising their manuscript to make it stronger. Translational knowledge on activity measures in human adults and especially children (since they do not move exactly the same way as adults) will bring more to the discussion in this manuscript. There are indeed challenges in collecting valid accelerometer data and there is a need to address that prior to using the outcome measure in dogs with chronic pain conditions such as osteoarthritis.

To make this submission even better, I’d like to present the following additional comments, questions and suggestions, which I believe will improve the manuscript.

Introduction

Line 46          Why abbreviation for green-lipped mussel?

Line 39-43    There is no reference for this section on delta-G.

Line 83-85    The aim needs to be clarified. Why: “delta-G”? Is the aim to evaluate this specific device, or to evaluate if delta-G measures of activity can be used to predict speed and gait?

Materials and methods

Line 93-94    “…enough data to validate the accelerometer using…” Was this the aim of this study? To validate the device? The accelerometer? Compared to the aim in line 83-85.

Line 102        “…a sampling rate of 10 Hz” Was the sampling rate 10 Hz? Not 100 Hz? Does the device have enough capacity, i.e. sampling rate, to measure triaxial acceleration in moderate to vigorous intensity (canter)?

Line 121        “…, and treadmill speed was recorded” Was this the speed used in the statistical models? Or was the speed of the dogs assessed in another way? How was the speed of the treadmill calibrated?

Line 150-160                      There is a lack of information to the reader. Please clarify further the methods around PCA. How was the PCA conducted and how exactly were the factor loadings being used by the authors? Please add a reference on PCA. Should the PCA description paragraph be moved to the statistics section?

Line 157        “The eigenvector values….” Do the authors mean the factor loadings? How many factors were extracted and by which rules? Please add to the presentation of the results and clarify.

Table 1          Indicates rather equal loading in some of the morphometric measurements while the loading on body length was low.  How exactly were the various loadings used as a variable “skeletal size”?

Results

Figure 3, Figure 4 and Figure 5:

Plateau or inverted u-shape phenomenon has been studied in accelerometer data from adult humans in moderate to vigorous intensity activities and in children. How does this phenomenon relate to the data presented in this study? What is the impact of reduced sampling rate in the device used in this study on the data points collected? I am not convinced that the accelerometer used in this study measures all triaxial accelerations in the dogs during faster gaits. And I am afraid that the results can be biased by the low sampling rate. Please elaborated on my concern about this and add to discussion.

Line 302        Add abbreviation for OA.

Line 303        Check reference system please.

Line 280-371 Discussion. I really like the discussion. Add to discussion: the elaboration on inverted u-shape phenomenon and the impact of reduced sampling rate in the accelerometer used in this study.

Line 374        Conclusion: I agree with the authors’ conclusion that this study demonstrated that delta g from this device can be used to separate dogs’ gaits into walk and a faster gait (trot/canter). But I am not convinced that there might not be a problem with the experimental protocol using this device. Please revise the conclusion accordingly.

Line 375        “… a faster gait (trot, canter, or gallop)…” Please remove gallop.

Simple summary

Line 14          “i.e. trot, canter, or gallop…” Please remove gallop.

Line 15-16    Please revise the conclusion accordingly.

Abstract

Line 27          “… and faster( trot, canter, or gallop…)..” Please remove gallop.

Line 30-32    Please revise the conclusion accordingly.

Author Response

The authors would like to thank the reviewer for the thorough and insightful comments.  We believe that the emendations have significantly improved the manuscript. 

Introduction

Line 46          Why abbreviation for green-lipped mussel?

Answer:  Thank you for pointing that out – the abbreviation was the jetsam of an earlier version, and has now been removed.

Line 39-43    There is no reference for this section on delta-G.

Answer:  We have added the reference Albright et al (2017) to support this section.

Albright, J.D.; Seddigh,i R.M.; Ng, Z.; Sun, X.; Rezac, D.J.; Effect of environmental noise and music on dexmedetomidine-induced sedation in dogs. PeerJ. 2017 Jul 31;5:e3659. doi: 10.7717/peerj.3659

Line 83-85    The aim needs to be clarified. Why: “delta-G”? Is the aim to evaluate this specific device, or to evaluate if delta-G measures of activity can be used to predict speed and gait?

Answer:  It was both, but it is argued that delta-g will correlate strongly across devices, even if the units are different, whereas the correlation of the activity count, and the ability to define finer actions such as gait, will not be closely correlating.

The section now reads: “Therefore, this study aimed to measure controlled activity in dogs using accelerometry and determine if the "delta-G" measures of activity (as opposed to a propriety specific activity count) could be reliably converted to estimates of speed and categorisation of gait.”

Materials and methods

Line 93-94    “…enough data to validate the accelerometer using…” Was this the aim of this study? To validate the device? The accelerometer? Compared to the aim in line 83-85.

Answer:  This has been changed to read “Therefore, it was estimated that eight dogs would provide enough data to validate the use of delta-G applying our proposed method”

Line 102        “…a sampling rate of 10 Hz” Was the sampling rate 10 Hz? Not 100 Hz? Does the device have enough capacity, i.e. sampling rate, to measure triaxial acceleration in moderate to vigorous intensity (canter)?

Answer:  That is correct – we used 10Hz.  As I suspect the reviewer is aware, the true sampling rate of the electronics at the most basic level is much higher than 10Hz.  However, higher sampling rates increase background noise, and create data sets that need to be smoothed.  Thus, summing the intervals to lower frequencies produces a compromise between too frequent noisy data that requires manipulation, and not frequent enough which removes the ability to detect movement at a frequency we desired.  As we discuss below, our observations and suspicions are that it is not the sampling rate that was limiting, indeed any greater rate almost certainly increased the noise to signal ratio, but rather it was the limitation of a collar mounted device and the way that dogs gait, that combined to prevent better categorisation of faster gaits.

Line 121        “…, and treadmill speed was recorded” Was this the speed used in the statistical models? Or was the speed of the dogs assessed in another way? How was the speed of the treadmill calibrated?

Answer:  The speed of the treadmill was calibrated by the manufacturer, shortly before study commencement.  In addition, the speed was checked at the time, using markers that were secured to the belt at a precisely measured distance apart, and their sequential movement was timed as they passed a static point at different speeds. 

This description has been added to the section, starting at line 112. 

Line 150-160                      There is a lack of information to the reader. Please clarify further the methods around PCA. How was the PCA conducted and how exactly were the factor loadings being used by the authors? Please add a reference on PCA. Should the PCA description paragraph be moved to the statistics section?

Answer:  We appreciate the reviewer’s desire to have more written in this paper.  The authors decided that since the procedure used was identical to the previously published method, that readers could be referred back to that particular paper, rather than re-write the methods here.

In answer to the reviewer – we have used PCA to produce a composite variable that we refer to as “skeletal size”.  It is unknown if a single skeletal measurement, such as forelimb length, might substitute for all of the variables including limb length: bodylength ratio, overall height, hindlimb conformation etc.  Thus, we have used the composite “skeletal size” measurement to standardise the size of dogs, whilst accounting for the difference in conformation.  As it happens, the approach is successful within a narrow breed range (i.e. those used in this study), but would need breed-specific values to be used more widely.

We have reworded the entire section for clarity.

 Line 157        “The eigenvector values….” Do the authors mean the factor loadings? How many factors were extracted and by which rules? Please add to the presentation of the results and clarify.

Answer:  Thank you for that correction – that should have read “factor loadings”.  The factors that were used were established in the paper Leung et al, and were head length, head circumference, foreleg length, hindleg length, body length, and thoracic girth.  In the previous paper, all measurements had similar loadings of the same sign (i.e. very similar vectors), and thus provided similar weighting to the overall variation in the composite variable skeletal size.  We have reworded the section to clarify. 

Table 1          Indicates rather equal loading in some of the morphometric measurements while the loading on body length was low.  How exactly were the various loadings used as a variable “skeletal size”?

Answer:  That is correct.  Calculation of the skeletal size of each dog using the equation

where  represents the values of the morphometric measurements a – f (head length, circumference etc) of the dog, and the remaining terms  represents the mean of the measurements,  represents the standard deviation of the measurements, and  represents the loadings of the measurements.

It was decided that including the full explanation of the methodology was inappropriate for this submission, and that reference back to the original study would suffice.  We hope that the expanded explanation and clearer reference back to the previous study will be acceptable.      

Results

Figure 3, Figure 4 and Figure 5:

Plateau or inverted u-shape phenomenon has been studied in accelerometer data from adult humans in moderate to vigorous intensity activities and in children. How does this phenomenon relate to the data presented in this study? What is the impact of reduced sampling rate in the device used in this study on the data points collected? I am not convinced that the accelerometer used in this study measures all triaxial accelerations in the dogs during faster gaits. And I am afraid that the results can be biased by the low sampling rate. Please elaborated on my concern about this and add to discussion.

Answer:  We do not believe that the plateau was caused by the sampling rate used.  As the reviewer notes, a similar distribution of data has been seen in other species.  We believe that the most likely explanation is that a collar-mounted accelerometer is unable to easily detect the increase in stride length that occurs as the speed within the canter gait increases.  Were it to be mounted on the legs, we posit that we would have detected a change.  In response to the reviewers questions, we have inserted the following paragraph into the discussion as requested:

“In humans running on a treadmill while wearing a triaxial accelerometer, there is an inverse curvilinear relationship between exercise intensity and activity counts, and in one study it was found that the activity count decreased at running speeds faster than c. 3m/sec (Miller et al).  In that study, the authors showed that the phenomenon was due to the limitations in the accelerometer, which was designed to detect accelerations with frequencies between 0.25 and 2.5 Hz, which are too slow, since acceleration during running at faster speeds exceeds 2.5 Hz in humans. 

The device used in our study had a sampling rate of 10Hz, but it is still possible that a higher sampling frequency could detect higher frequency accelerations and allow a better measurement of speed beyond 3 m/sec in dogs.  However, we posit that mounting the accelerometer on the collar limits the ability to detect those accelerations, and that stride length is the gait change responsible for increasing speeds during a canter.  Thus, attachment of an accelerometer to the legs of a dog, whilst sampling at a rate of 10Hz, is more likely to be accurate at measuring speed.  Nonetheless, we wanted to evaluate the more convenient collar-mounted approach for this study.  Further studies are required to determine whether a higher sampling frequency, or repositioning of accelerometers will give a better measure of speed.”

John D, Miller R, Kozey-Keadle S, Caldwell G, Freedson P. Biomechanical examination of the 'plateau phenomenon' in ActiGraph vertical activity counts. Physiol Meas. 2012;33(2):219-230. doi:10.1088/0967-3334/33/2/219

Line 302        Add abbreviation for OA.

Answer:  We have replaced that with the abbreviation.

Line 303        Check reference system please.

Answer:  Apologies – that has been corrected now.

Line 280-371 Discussion. I really like the discussion. Add to discussion: the elaboration on inverted u-shape phenomenon and the impact of reduced sampling rate in the accelerometer used in this study.

Answer:  We appreciate this suggestion by the reviewer, and we have added to this discussion as noted above.

Line 374        Conclusion: I agree with the authors’ conclusion that this study demonstrated that delta g from this device can be used to separate dogs’ gaits into walk and a faster gait (trot/canter). But I am not convinced that there might not be a problem with the experimental protocol using this device. Please revise the conclusion accordingly.

We have reworded the conclusion, to include:

“but estimation of speed from the accelerometer signal above 3m/s was inaccurate.  We were unable to determine if that inaccuracy was due to the sampling frequency of the particular device used, or if that is an inherent limitation of collar-mounted accelerometers in dogs”

Line 375        “… a faster gait (trot, canter, or gallop)…” Please remove gallop.

Answer:  Thank you for that correction, we have removed the word.

Simple summary

Line 14          “i.e. trot, canter, or gallop…” Please remove gallop.

As above.

Line 15-16    Please revise the conclusion accordingly. 

Abstract

Line 27          “… and faster( trot, canter, or gallop…)..” Please remove gallop.

As above

Line 30-32    Please revise the conclusion accordingly.

As above